# Immunotherapy Targeting PD-1/PD-L1 in Early-Stage Triple-Negative Breast Cancer

**DOI:** 10.3390/jpm13030526

**Published:** 2023-03-15

**Authors:** Tinglin Yang, Wenhui Li, Tao Huang, Jun Zhou

**Affiliations:** Department of Breast and Thyroid Surgery, Union Hospital, Tongji Medical College, Huazhong University of Science and Technology, Wuhan 430022, China

**Keywords:** early-stage breast cancer, triple-negative breast cancer, immunotherapy, PD-1/PD-L1, irAEs

## Abstract

The advent of immunotherapy, especially immune checkpoint inhibitors (ICIs), has revolutionized antitumor therapy. Programmed cell death receptor 1 (PD-1) and programmed cell death ligand 1 (PD-L1) are among the most promising targets for encouraging the immune system to eliminate cancer cells. PD-1/PD-L1 have made clinical remission for numerous solid tumors, including metastatic triple-negative breast cancer (TNBC). In recent years, integrating PD-1/PD-L1 inhibitors into existing treatments in early-stage TNBC has attracted wide attention. Herein, we summarize the clinical benefit of PD-1/PD-L1 inhibitors plus neoadjuvant chemotherapy, adjuvant chemotherapy, and targeted therapy in early-stage TNBC. Possible immunotherapy biomarkers, immune-related adverse events (irAEs), and the key challenges faced in TNBC anti-PD-1/PD-L1 therapy are also concluded. Numerous studies on immunotherapy are ongoing, and PD-1/PD-L1 inhibitors have demonstrated great clinical prospects in early-stage TNBC. To maximize the efficacy of anti-PD-1/PD-L1 therapy, further research into the challenges which still exist is necessary.

## 1. Introduction

Breast cancer exhibits the top incidence rate of all malignancies, ranking as the chief cause of cancer mortality in women [1]. Among all breast cancers, the subtype triple-negative breast cancer (TNBC) makes up 15 to 20%, which is defined as estrogen receptor (ER)-negative, progesterone receptor (PR)-negative, and human epidermal growth factor receptor-2 (HER-2)-negative [2]. TNBC is well characterized by a poorer prognosis, higher possibility for relapse, and earlier age of onset than other subtypes [3,4]. Therefore, it is imperative to explore novel and efficacious remedies for TNBC.

Recent progress in this regard has been achieved in the utilization of immunotherapy aimed at enhancing antitumor immunity to eliminate malignant cells, which has been deemed as a significant breakthrough in revolutionizing antitumor therapy. As a major category of immunotherapy, immune checkpoint inhibitors (ICIs) are now entering extensive clinical practice. Tumor cells can escape immune surveillance by exploiting immune checkpoints, which contribute to activating coinhibitory signaling pathways and immune tolerance [5]. Programmed cell death receptor 1 (PD-1) and programmed cell death ligand 1 (PD-L1) are currently among the most promising ICI targets to be blocked to attack malignant cells through an immune-mediated process [6,7]. 

PD-1/PD-L1 inhibitors have shown substantial clinical benefits in malignancies that occur in lung, kidney, bladder, and skin [8,9,10,11]. Compared with other subtypes, considerable evidence has shown that immunotherapy may have a better response rate in TNBC. There exist more tumor-infiltrating lymphocytes (TILs), larger numbers of mutations, and relatively higher PD-L1 expression in TNBC [12,13]. Additionally, higher levels of TILs indicate better outcomes in TNBC [14]. ICIs can strengthen the progress of immune clearance, thus making the use of PD-1/PD-L1 inhibitors a possible strategy against TNBC [15]. 

Monumental progress has been seen in the field of combining ICIs with adjuvant chemotherapy in metastatic TNBC, although there are some limitations in monotherapy with PD-1/PD-L1 inhibitors [16,17]. In 2019, the European Commission and the Food and Drug Administration (FDA) approved atezolizumab combined with nab-paclitaxel in metastatic TNBC with positive PD-L1, which established the initial immunotherapy regimen approved for breast cancer patients [18]. Inspiring results in metastatic TNBC greatly boosted the subsequent investigation into the usage of PD-1/PD-L1 monoclonal antibodies in early-stage TNBC clinical practice, and more encouraging data has recently emerged. Although previous reviews and meta-analyses have provided insights into anti-PD-1/PD-L1 therapy in cancer, the efficacy of PD-1/PD-L1 inhibitors, in early-stage TNBC in particular, has never been systematically reviewed [19,20,21]. In this article, immunotherapies based on PD-1/PD-L1 inhibitors in early-stage TNBC are summarized, including neoadjuvant chemotherapy, adjuvant chemotherapy, and targeted therapy. The latest results from clinical trials are summarized, as well as possible immunotherapy biomarkers, immune-related adverse events (irAEs), and the challenges which are being faced in the field. To succeed in the application of PD-1/PD-L1 inhibitors in early-stage TNBC, efforts in both basic research and clinical development are needed. 

## 2. The Rationale for PD-1/PD-L1 Blockade

PD-1 and PD-L1 are demonstrated as members of the immunoglobulin (Ig) superfamily, and both have been identified as transmembrane proteins. PD-1 can be detected on the activated T-cell membrane surface [22,23]. The presence of PD-L1 in normal tissue is well-documented, acting as the ligand of PD-1. T-cell activity is inhibited by PD-1 and PD-L1 interactions, resulting in immune tolerance. The PD-1/PD-L1 pathway is a crucial element of physiological immune homeostasis [24]. However, abnormally expressed PD-L1 has been detected in multiple kinds of carcinoma, including breast cancer, colorectal cancer, lung cancer, and melanoma [25]. It may have connections with cytokines in the tumor microenvironment (TME), especially interferon-γ (IFN-γ). When undergoing an immune attack, IFN-γ is the prominent soluble cytokine inducing the expression of PD-L1 in tumor cells [26]. As IFN-γ binds to its receptor, the enhanced expression of transcription factors upregulates PD-L1 transcription and translation in cancer cells [27]. 

The proliferation of lymphocytes mediated by the T-cell receptor (TCR) is suppressed when PD-1 and PD-L1 are engaged, hence inducing immunosurveillance [28,29]. The higher the expression of PD-L1 in tumors, the more immune-suppressive the TME may become, as shown in Figure 1. Inhibiting PD-1 or PD-L1 can contribute to antitumor immunity and induce carcinoma regression by various pathways, including (1) reinvigorating lymphocyte activity and cytotoxic cytokine release; (2) activating and proliferating CD8+ T-cells specific to tumor antigens; (3) dislodging the apoptosis of lymphocytes induced by PD-1/PD-L1 interaction; and (4) boosting the immune discrimination of tumor cells [30,31]. 

In the new era of personalized medicine, PD-1/PD-L1 inhibitors are attracting ever-increasing attention. PD-1 antibodies (i.e., pembrolizumab and nivolumab) and PD-L1 antibodies (i.e., atezolizumab, durvalumab, and avelumab) have generated effective efficacy in multiple kinds of malignancies [8,9,10,11,18]. Following the permission of atezolizumab in metastatic TNBC therapy, integrating PD-1/PD-L1 inhibitors into early-stage TNBC treatment to improve prognosis has entered the research spotlight. In recent years, a number of clinical trials have been conducted to evaluate the clinical profit of PD-1/PD-L1 inhibitors in early-stage TNBC. The major trend in current trials is to combine anti-PD-1/PD-L1 therapy with neoadjuvant chemotherapy, adjuvant therapy, or targeted therapy.

## 3. PD-1/PD-L1 Inhibitors Plus Neoadjuvant Chemotherapy

Neoadjuvant therapy is the most recommended strategy for treating high-risk, early-stage TNBC [32,33]. It has also been demonstrated that ICIs plus neoadjuvant chemotherapy may be promising in treating early-stage TNBC, and nine representative studies in this aspect are shown in Table 1. For participants receiving neoadjuvant chemotherapy, not only is the pathological complete response (pCR) rate estimated as an observation endpoint, but some survival and safety indicators are also included.

The GeparNuevo study (NCT02685059) is a phase II trial with a placebo control. It consists of 174 primary TNBC patients in clinical stage I, II, or III. The participants were treated with nab-paclitaxel, epirubicin and cyclophosphamide, plus durvalumab or placebo. Although the patients receiving durvalumab witnessed a slight increase in their pCR rate compared with the patients who were receiving the placebo, there was no statistical significance (53.4% versus [vs.] 44.2%, *p* = 0.287). Favorably, there was a statistically significant tendency for improved three-year survival, including invasive disease-free survival (iDFS), distant disease-free survival (DDFS), overall survival (OS), and no new safety signals occurring. In the durvalumab group and placebo group, iDFS was 85.6% vs. 77.2% (*p* = 0.036), DDFS was 91.7% vs. 78.4% (*p* = 0.005), and OS was 95.2% vs. 83.5% (*p* = 0.006), respectively [34,35,36].

In the phase II clinical trial I-SPY2 (NCT01042379), a subgroup was designed to investigate pembrolizumab with neoadjuvant chemotherapy. In total, 69 subjects with early-stage breast cancer were randomized to different treatment groups of pembrolizumab combined with paclitaxel, adriamycin, and cyclophosphamide, while 181 patients who accepted standard neoadjuvant chemotherapy were taken as the control group. For the pembrolizumab vs. control arm in the total population, the pCR rate was 44% vs. 17%. A pCR rate of 60% was achieved with pembrolizumab compared with 22% with placebo. In those at high risk in early-stage TNBC, pembrolizumab exceeded the estimated pCR rates by more than twice, indicating a promising clinical future for pembrolizumab [37].

The NeoPACT phase II trial (NCT03639948) aimed to investigate neoadjuvant pembrolizumab with carboplatin and docetaxel in 117 TNBC patients. The study measured pCR rates as a primary endpoint, while event-free survival (EFS) rates and residual tumor burden (RCB) were taken as the secondary endpoints. No patient experienced disease progression during the neoadjuvant therapy. The pCR rate of pembrolizumab plus carboplatin and docetaxel neoadjuvant therapy was 58%, which is comparable to the pCR rate seen in neoadjuvant immunotherapy with anthracycline-based chemotherapy. The 2-year EFS rate was 89% for the total population, 98% for the pCR group, and 82% for the non-pCR group. These results support that it is not inferior to integrate pembrolizumab with a non-anthracycline neoadjuvant chemotherapy strategy, which may inspire new clinical trials in early-stage TNBC [38]. 

The KEYNOTE-173 phase Ib trial (NCT02622074) is a multicohort study which evaluated six regimens of pembrolizumab with chemotherapy as neoadjuvant treatment. In total, 60 early-stage TNBC patients were registered to six cohorts in the trial, and there was a range of pCR rates from 49% to 71%. The overall pCR rate ended at 60%, and a range of 80% to 100% 12-month EFS and OS rates across all cohorts (100% for four cohorts) was observed. The study demonstrated the promising antitumor activity of pembrolizumab plus neoadjuvant chemotherapy, and a positive connection was proven between PD-L1 expression, levels of stromal TILs (sTILs), and pCR rates [39]. 

On the basis of the KEYNOTE-173 study, the phase III KEYNOTE-522 trial (NCT03036488) was conducted. This is a study of neoadjuvant and adjuvant pembrolizumab with chemotherapy in early-stage TNBC patients. In total 1174 patients who were previously untreated, non-metastatic, and centrally confirmed with TNBC were recruited and randomized to the pembrolizumab group and control group in a 2:1 ratio. The patients in the experiment group were administered pembrolizumab, with paclitaxel and carboplatin, followed by pembrolizumab for four cycles plus anthracyclines as neoadjuvant therapy. Following definitive surgery, nine cycles of pembrolizumab and chemotherapy were administrated as an adjuvant treatment. For patients in the control group, pembrolizumab was replaced by a placebo. The dual primary endpoints included pCR and EFS. In the primary analysis, the pCR rates were 64.8% and 51.2% of the experiment group and the control group (*p* = 0.00055). The interim analysis also revealed that pembrolizumab with chemotherapy could significantly increase pCR rates regardless of the PD-L1 status expressed by the breast tumor. PD-L1-positive patients had a 14.2% increase in pCR (68.9% vs. 54.9%, 95% confidence interval [CI]: 5.3% to 23.1%), and PD-L1-negative patients had an 18.3% increase (45.3% vs. 30.3%, 95% CI: −3.3% to 36.8%). An analysis of the subgroups demonstrated that pembrolizumab is more likely to benefit patients with a heavier tumor burden, in a late stage of the disease, and with positive lymph nodes. In the fourth interim analysis, the estimated 36-month EFS rate in the experiment group was 84.5%, while it was 76.8% in the control group. Previous studies have confirmed that RCB grades after neoadjuvant chemotherapy could be used as a prognosis biomarker [40,41,42]. The KEYNOTE-522 study analyzed the correlation between RCB grades and EFS, which was reported at the American Society of Clinical Oncology (ASCO) meeting in 2022. The study suggested that the addition of immunotherapy reduced RCB scores and improved EFS. Based on a series of promising results and the relatively mild side effects of pembrolizumab from KEYNOTE-522 and other studies, both the European Medicines Agency (EMA) and the FDA approved joining pembrolizumab and chemotherapy for neoadjuvant therapy and subsequent adjuvant therapy, establishing the first immunotherapy regimen approved in high-risk, early-stage TNBC [36,43]. 

The NeoTRIPaPDL1 phase III trial (NCT02620280) adopted the approach of integrating atezolizumab with carboplatin and nab-paclitaxel. In total, 280 early-stage TNBC patients were recruited and assigned randomly to undergo chemotherapy with or without atezolizumab as a neoadjuvant therapy. The trial aimed to compare EFS, as well as the rate of pCR. In the intention-to-treat (ITT) population, the pCR rate increased by 2.7% (43.5% vs. 40.8%) in the experiment group, showing no statistical significance (*p* = 0.066), which doubted immunotherapy in early-stage TNBC patients. The study also defined a group of “immune-rich” patients who had higher PD-L1 expressions and more TILs in the TME. Positively, the 2021 European Society for Medical Oncology (ESMO) congress reported that the pCR rate in “immune-rich” patients increased (87% vs. 72%) with the atezolizumab treatment. A long-term follow-up on EFS is necessary [44].

Another phase III study, IMpassion 031 (NCT03197935), aimed to assess the addition of atezolizumab to neoadjuvant chemotherapy in 333 early-stage TNBC patients. Patients were enrolled and equally randomized to atezolizumab and placebo groups. The primary endpoints were pCR rates among ITT and PD-L1-positive patients, while the secondary endpoints aimed to estimate EFS, DFS, and OS. There was a dramatic improvement in pCR rates in atezolizumab-containing groups, both in the ITT population (58% vs. 41%, *p* = 0.0044) and in PD-L1-positive participants (69% vs. 49%, *p* = 0.021). The existing results consolidated that atezolizumab combined with nab-paclitaxel and anthracycline can potentially enhance clinical benefit in neoadjuvant therapy for TNBC [45]. 

In addition, there are several ongoing trials exploring atezolizumab as a neoadjuvant therapy in early-stage TNBC. Atezolizumab combined with neoadjuvant chemotherapy and subsequent adjuvant atezolizumab for one year is being evaluated in the NSABP B-59 phase III study (NCT03281954). The trial allocated 1550 patients with stage T2 or T3 TNBC and randomized patients to the atezolizumab group or the placebo group. Patients in the atezolizumab group are receiving atezolizumab plus paclitaxel and carboplatin, followed by anthracycline. The placebo is replacing atezolizumab in the control group. EFS will be taken as the primary endpoint, and the secondary endpoints will be OS, the pCR rate, and DDFS [46]. Another ongoing phase II study MIRINAE (NCT03756298) aims to assess the clinical benefit of joining atezolizumab with capecitabine in TNBC patients who have completed neoadjuvant treatment but still present with residual tumors. In this study, 284 patients have been recruited to participate, and the major outcome is five-year disease-free survival (DFS). We are expecting favorable results to emerge from this study [47].

## 4. PD-1/PD-L1 Inhibitors Plus Adjuvant Chemotherapy

As a PD-1 inhibitor, pembrolizumab in early-stage, high-risk TNBC neoadjuvant therapy has been endorsed by the FDA. However, it remains unclear how immunotherapy could benefit early-stage TNBC patients. Clinical trials integrating PD-1/PD-L1 antibodies with adjuvant therapy are ongoing, as displayed in Table 2. 

The IMpassion 030 trial (NCT03498716) is a prospective phase III study which aims to evaluate whether adjuvant atezolizumab in conjunction with anthracycline/paclitaxel chemotherapy is clinically beneficial for operable TNBC patients. The study was designed to enroll 2300 stage II or III TNBC patients. Patients are set to undergo atezolizumab and chemotherapy, or chemotherapy alone. iDFS has been set as the primary endpoint, while OS, DFS, recurrence-free interval (RFI), distant RFI, and adverse events are to be the secondary endpoints. An investigation and extensive analysis are underway to determine the results [48]. 

In the A-BRAVE trial (NCT02926196), avelumab is being assessed as an adjuvant treatment and post-neoadjuvant treatment. In this phase III study, 474 participants who have non-metastatic primary invasive high-risk TNBC were assigned randomly to the avelumab arm and the observation arm. Patients will be administered 200 mg of avelumab or be observed under the guidelines. DFS in the total population and DFS in PD-L1-positive patients were designed to be primary outcome measurements, while OS and safety profiles will be analyzed as the secondary outcome measurements [49]. 

Another phase III trial (NCT02954874) is investigating pembrolizumab as an adjuvant treatment for TNBC in the Southwest Oncology Group (SWOG 1418). A total of 1155 patients who have positive lymph nodes or residual tumors after neoadjuvant chemotherapy have been recruited, and all patients underwent their final breast surgery before being registered. In the observation arm, patients will be monitored at standard clinical intervals with no immunotherapy. In the pembrolizumab arm, patients will receive pembrolizumab. The primary endpoints are iDFS, the severity of fatigue, and physical function, and the secondary endpoints are OS, distant DFS, adverse events, etc.

More research into immunotherapy as an adjuvant treatment is required in order to clarify its clinical outlook.

## 5. PD-1/PD-L1 Inhibitors Plus Targeted Therapy

Following the in-depth understanding of gene mutations and molecular pathways in breast cancer, therapies targeting certain molecules make up a significant part of TNBC treatment. Various kinds of targeted therapies may help to stimulate antitumor immunity, hence making them potential strategies to rise above the innate resistance to PD-1/PD-L1 monoclonal antibodies [50]. Research on combining poly (ADP-ribose) polymerase (PARP) inhibitors, Akt inhibitors, vascular endothelial growth factor receptor (VEGFR) inhibitors, or cell cyclin-dependent kinases 4 and 6 (CDK4/6) inhibitors with PD-1/PD-L1 antibodies in early-stage TNBC is already underway, as displayed in Table 3. 

It has been shown in preclinical studies that PARP inhibition promotes neoantigen presentation and activates T-cells. As the proportion of TILs and PD-L1 expression are also upregulated by PARP inhibitors, PARP inhibitors are assumed to treat TNBC with PD-1/PD-L1 inhibitors [51,52]. Accessing the integration of ICIs and PARP inhibitors as a neoadjuvant setting was carried out by a subgroup in the I-SPY2 study (NCT01042379). In total, 372 high-risk patients with HER2-negative breast cancer were investigated, and 163 TNBC patients were included. Twenty-one TNBC patients were randomized to the experimental group, who received durvalumab, the PARP inhibitor olaparib, paclitaxel, and adriamycin/cyclophosphamide. The other 142 TNBC patients in the control group only underwent chemotherapy. Compared with the control group, the pCR rate of the experiment group nearly doubled (37% vs. 20%) in all participants. The extensive analysis of the TNBC subgroup revealed that the pCR rates were 47% vs. 27%, respectively. The results of this study further revealed that patients with TNBC can benefit from the combination of immunosuppressants and PARP inhibitors [53]. A window-of-opportunity clinical trial (NCT03594396) has recruited 54 participants with stage II/III TNBC; olaparib and durvalumab will be administered to patients before commencing standard neoadjuvant chemotherapy. The changes in tumor biology will be compared and analyzed as the primary endpoint, while the pCR rates, response rate, and adverse events are being collected as the secondary endpoints. 

Preclinical studies have proven that the PI3K/Akt signaling pathway has a tough relationship with CD8+ T-cell differentiation and memory activity [54,55]. Inhibiting Akt is a promising way to regulate the immunosuppressive TME by reviving memory T-cells [56]. There are reasons to expect that combining PD-1/PD-L1 antibodies and Akt inhibitors may benefit breast cancer patients. The BARBICAN trial (NCT05498896) was organized to evaluate the addition of the Akt inhibitor ipatasertib to chemotherapy and atezolizumab in early-stage TNBC patients with and without PI3CA/Akt1/PTEN genetic mutations. In total, 146 TNBC patients were randomly assigned to the experimental group and control group. The participants received atezolizumab with paclitaxel, doxorubicin, and cyclophosphamide in the control group. In the experimental group, ipatasertib was administered to atezolizumab and chemotherapy. The pCR rates and five-year objective response rates (ORR) were set as the primary and secondary endpoints, respectively. In all patients, the pCR rates were 48.5% for the control group and 49.3% for the experiment group (*p* = 0.729). Congruously, the benefit of Akt inhibition was observed neither in PD-L1-positive nor PD-L1-negative subgroups [57]. 

The vascular endothelial growth factor (VEGF) is secreted by immune cells in the TME, including tumor-associated macrophages (TAM). Besides contributing to angiogenesis, the VEGF can act as an immunomodulator to promote local and systemic immunosuppressive TME. VEGFs can result in the suppression of antigen presentation, the stimulation of regulatory T-cells, and the activation of tumor-associated macrophages. Therefore, VEGFs can be considered immune suppressive [58,59]. A preclinical study proved that anti-angiogenic therapy can sensitize TNBC cell lines to PD-1 blockade in vitro [60]. As a novel treatment approach for cancer, combining targeting VEGFR with immunotherapy is set to enter clinical trials. The BRE-03 trial (NCT04427293) is a phase I window-of-opportunity trial which aims to enroll 12 patients with stage I, II, or III TNBC. Lenvatinib, a VEGFR inhibitor, will be administrated for a week, and 200 mg of pembrolizumab will be administered on the first day of the treatment. Infiltration of CD8+ TILs in primary tumors, representing a T-cell inflamed TME, will be measured as the primary outcome. Another phase II trial in China called NeoCAT aims to recruit 58 patients with operable invasive TNBC (T1cN1-2 or T2-4N0-2) and high proportions of TILs (>10% in baseline breast tumor). Eligible patients will receive the PD-1 inhibitor camrelizumab and the VEGFR inhibitor apatinib as neoadjuvant treatments. After eight cycles, the pCR rate will be analyzed as the primary endpoint. Secondary outcome measurements include ORR, breast conservation rate, emergent adverse events, etc.

CDK4/6 are the key factors in controlling the cell cycle, which can block the phosphorylation of retinoblastoma inhibitors and cause cell cycle arrest [61]. Studies have shown that CDK4/6 is correlated to the onset and progression of a variety of malignancies [62]. Inhibitors of CDK4/6 have demonstrated initial success in treating hormone receptor-positive breast cancer and a diversity of malignancies [63,64]. The treatment with CDK4/6 inhibitors activates CD8+ T-cells and heightens the effect of anti-PD-1/PD-L1 agents, indicating the theoretical possibility of the addition of CDK4/6 inhibitors to PD-1/PD-L1 antibodies [65,66]. In a phase II single-arm study (NCT05112536), the CDK4/6 inhibitor trilaciclib combined with chemotherapy and pembrolizumab will be evaluated for its mechanism of action, safety, and efficacy. A total of 24 patients with TNBC will be administered trilaciclib, pembrolizumab, anthracycline, and paclitaxel. After taking a single dose of trilaciclib for seven days as the lead-in, the primary outcome will be the shift in the ratio of CD8+ T-cells to regulatory T-cells (Tregs) in cancer. The pCR rates and emergent adverse events are set as the secondary endpoints.

Despite the fact that anti-PD-1/PD-L1 agents in conjunction with targeted therapy need to be explored further in early-stage TNBC, clinical research on this topic already shows promising prospects.

## 6. Potential Biomarkers of Immunotherapy Response in TNBC

Despite the progress made in anti-PD-1/PD-L1 treatment, some patients still show no response to immunotherapy regimens. The appropriate biomarkers of immunotherapy response can be a significant basis for initiating ICIs treatment, which has been under exploration in TNBC clinical trials for many years. Emerging potential biomarkers include PD-L1 expression, tumor mutational burden (TMB), TILs, and mismatch repair (MMR) deficiency [67,68].

In previous clinical studies in metastatic TNBC, immunotherapy was more likely to work for PD-L1-positive patients, as discovered by the KEYNOTE-012 study and IMpassion 130 study [18,69]. Contradictorily, it is still possible for PD-L1-negative patients to show sensitivity to ICIs. In the KEYNOTE-522 and the NeoTRIPaPDL1 studies, response to ICIs showed little to no correlation with PD-L1 expression [43,44]. To make matters worse, there is currently no standard assay to detect PD-L1. Commonly used immunohistochemical assays include SP263, DAKO 22C3, 28–8, Ventana SP142, and 73–10 assays. Adopting the different expression-scoring criteria of PD-L1 in different clinical trials leads to different conclusions [70,71]. As a result, it will be suggested that the patient receives atezolizumab plus nab-paclitaxel therapy if the patient is PD-L1 positive in the SP142 assay (score ≥ 1%). However, if one is PD-L1 positive in the 22C3 test (combined positive score [CPS] ≥ 10), it would be suggested that the patient receives pembrolizumab therapy [72,73,74]. Owing to this, the function of PD-L1 expression as an immunotherapy biomarker in TNBC is still under debate. Moreover, a few PD-L1-negative patients do indeed benefit from PD-1/PD-L1 blockade. Because of this finding, there is a need to screen more biomarkers.

In tumors, TMB denotes how many somatic mutations there are in gene-coding regions. Elevated TMB contributes to more neoantigens and triggers an intrinsic immune response as it is recognized by the immune system. In solid tumors, high TMB refers to > 10 mutations per megabase of DNA (mut/Mb), and it is connected with sensitivity to anti-PD-1/L1 agents across multiple malignancies [75,76]. Higher TMB also correlates with longer progression-free survival (PFS) in metastatic TNBC patients treated with ICIs [77]. The prospect of using TMB as an immunotherapeutic efficacy biomarker is promising.

Higher TILs levels have also shown an association with improved responses to ICIs. In the KEYNOTE-119 trial in metastatic TNBC, patients who had high TILs ended with an improved pembrolizumab response and survival [78]. Similarly, researchers found that TNBC patients in the early stage who had high CD8+ T-cell density and a high expression of immune-related genes concluded their treatment with higher pCR rates of durvalumab in the neoadjuvant setting [14,79]. PD-L1-positive patients showed longer PFS or OS if their tumors were infiltrated with high TILs in the IMpassion 130 trial [80]. The conjunction of multiple indicators is important, as this trial confirmed. 

The MMR system makes up a crucial part in sustaining the steadiness of the genome, which is divided into two categories: deficiency and proficiency. A deficiency of MMR leads to many somatic mutations occurring in simple repetitive sequences, resulting in microsatellite instability (MSI) [81]. It is more likely for MSI tumors to benefit from anti-PD-1/PD-L1 therapy [82]. For inoperable or metastatic solid carcinoma patients with elevated MSI, pembrolizumab was authorized by the FDA in May 2017. Nonetheless, dMMR in breast cancer was detected at only less than 2%, and the value of dMMR for PD-1/PD-L1 blockade therapy in TNBC still requires further investigation [83]. Taken together, immunotherapy biomarkers indicating prognosis in TNBC require further exploration, and the combination of different markers is a possible approach which may be successful.

## 7. IrAEs

IrAEs may exist in multiple organs in patients receiving anti-PD-1/PD-L1 treatment. The pathophysiology and molecular mechanism of irAEs are not yet completely understood. It was theoretically proven that irAEs may be caused by complex interactions between multiple factors including T-cells, antibodies, and cytokines that are autoreactive [84]. For instance, there is T-cell infiltration in both tumor and normal tissue. Activated T-cell upregulates the release of inflammatory cytokines, and the antigens shared by cancer and normal tissue can contribute to the progress of irAEs [85]. 

IrAEs vary from the targets of immunotherapy [86]. An onset of irAEs can arise at any time after initiating immunotherapy, in the range of weeks to months [87]. For patients who receive PD-1/PD-L1 antibodies, the observed adverse effects are involved mainly in the endocrine system, the digestive system, and the respiratory system. In addition, dermatologic toxicity and rare immune-related adverse events have also been reported [88]. 

The most common endocrine toxicities resulting from PD-1/PD-L1 inhibitors are hypothyroidism, hyperthyroidism, thyroiditis, hypophysitis, primary adrenal insufficiency, and insulin-dependent diabetes mellitus [89]. Hyperthyroidism occurs more frequently, which is often unabiding and may occur at the head of hypothyroidism. In TNBC studies using ICIs, the reported incidence of hypothyroidism ranged from 4% to 18.0%, whereas hyperthyroidism ranged from 1% to 4% [90,91]. An insufficiency in the adrenal gland can either be primary or secondary, and secondary adrenal insufficiency is known as hypophysitis. In both the ISPY-2 trial and the KEYNOTE-522 trial, there were participants who experienced adrenal insufficiency. Diabetes mellitus occurs at a relatively low frequency, at 1–2% across ICIs regimens [92]. In most endocrine toxicity cases, immunotherapy can be continued and high-dose corticoids are rarely needed. Meanwhile, a lifelong replacement is usually required for persisting endocrine deficiency [93]. 

Dermatologic toxicity is another irAE in TNBC immunotherapy, which has diverse clinical manifestations. Maculopapular rashes are believed to be the most frequently occurring. Dermatologic toxicity is rarely severe and usually does not discontinue ICIs therapy [94]. Hepatitis often presents as abnormal transaminases, occurring in 1–6.3% of TNBC participants during ICIs therapy [95]. The discontinuation of PD-1/PD-L1 antibodies ought to be taken into account when alanine transaminase or aspartate aminotransferase reaches four times more than the upper limit of what is deemed normal [93,96]. Gastrointestinal toxicity is often presented with diarrhea and colitis. In patients administrated with PD-1/PD-L1 inhibitors, diarrhea is more common than colitis [97]. Pneumonitis is a fatal pulmonary toxicity, with diverse clinical features and imaging findings. Pneumonitis can happen later than other irAEs after months of therapy initiation. There is a 1% to 4% incidence of immune-related pneumonitis among breast cancer patients [91,98]. Other rare irAEs have been reported, including toxicity towards the nervous system, heart, kidney, oculus, and hematopoiesis [99]. 

The clinical features of irAEs exhibit diversification among individuals and multiple systems are involved. Despite the fact that most irAEs can be treated by temporarily suppressing the immune system with corticoids or other immunosuppressant agents, severe irAEs lead to irreversible tissue damage, the discontinuation of treatment, or even death [100]. To manage irAEs, a deeper understanding is needed to optimize curative effects and prognosis.

## 8. Challenges in TNBC Immunotherapy

Up until now, PD-1/PD-L1 inhibitors have been suggested as a second-line therapy for PD-L1-positive TNBC patients. To further promote the progress of anti-PD-1/PD-L1 therapy in the quest for an improved prognosis, we should be acutely aware of the challenges that we face.

The mechanisms of primary and secondary immune escape to ICIs remain unclear. Some PD-L1-positive patients failed to respond to PD-1/PD-L1 inhibitors, referring to a primary immune escape. After responding to ICIs for years, patients may suffer from secondary immune escape, which results in tumor regrowth [101]. There is currently no complete understanding of the molecular drivers involved in the immune escape, however, they are considered to be crucial parts of resistance to immunotherapy. Multiple studies have outlined a variety of pathways that may lead to primary immune escape, including the transforming growth factor-β (TGFβ) signaling pathway, the Wnt-β-catenin signaling pathway, and the VEGF/VEGFR axis [102,103]. Ongoing clinical trials intend to combine targeted therapy with anti-PD-1/PD-L1 agents to overcome immune escape to ICIs, which may stimulate the immunogenicity of tumors.

In order to be an indication for a targeted population, biomarkers require further optimization. A number of trials have been conducted to analyze biomarkers that can be utilized to maximize personal benefits. However, immunotherapy biomarkers can perform differently under different circumstances. Although the 22C3 test of PD-L1 expression has been included in the National Comprehensive Cancer Network (NCCN) guidelines, some PD-L1-negative patients still respond to PD-1/PD-L1 inhibitors [104,105]. Moreover, the molecular subtypes of TNBC are under debate, indicating the heterogeneity of TNBC [106]. Composite biomarkers serving as the indications for anti-PD-1/PD-L1 therapy in TNBC patients require further research.

The pattern of combined regimens is another challenge. The response rate of immunotherapy monotherapy is somewhat low, which leads to the demand for combined regimens. Ongoing clinical trials are attempting to combine immunotherapy with existing antitumor therapies, including chemotherapy, targeted therapy, etc. In order to find the most effective remedy, combinations of various agents and immunotherapy are being explored. The NeoPACT trial proved the non-inferiority of non-anthracycline chemotherapy compared to anthracycline chemotherapy in combination with PD-L1 inhibitors [38]. Numerous studies on combining targeted therapy with immunotherapy are currently in progress. To optimize regimens, cumulative evidence is vital.

Improved endpoints are required to better evaluate clinical efficacy. Due to the fact that patients undergoing immunotherapy can exhibit flexible responses, clinical trials have been conducted with endpoints that are not well-matched for measuring curative effects in some cases. For instance, it is possible that patients may experience inflammation-induced tumor growth before they witness a reduction in tumor volume, which is called pseudoprogression [107]. The immune response evaluation criteria in solid tumors (iRECIST) was proposed in 2017, followed by the raise of immune-modified RECIST and immune-modified PFS in 2018 [108,109]. Unfortunately, the criteria are subject to limitations due to the variety of ICIs responses and the difficulty of distinguishing irAEs from tumor progression. An improved evaluation of the endpoints and the time points of follow-up require further discussion.

## 9. Conclusions

TNBC is deemed as the most challenging subtype of breast cancer, and inhibitors of PD-1/PD-L1 can favor TNBC patient outcomes by remodeling the TME and improving anti-tumor immunity. The KEYNOTE-522 trial established the first anti-PD-1/PD-L1 regimen permitted by the FDA in early-stage TNBC, which shed light on the utilization of PD-1/PD-L1 antibodies in the neoadjuvant setting. In addition, integrating anti-PD-1/PD-L1 regents into adjuvant chemotherapy and targeted therapy is showing promising therapeutic potential. As for the side effects, irAEs in early-stage TNBC immunotherapy should be monitored judiciously. Meanwhile, future work focusing on the challenges in the field is necessary to improve immunotherapy. Altogether, combining solid preclinical evidence and sufficient translational studies with convincing clinical trials is a firm foundation for accelerating the clinical development of PD-1/PD-L1 inhibitors in early-stage TNBC.

## Figures and Tables

**Figure 1 jpm-13-00526-f001:**
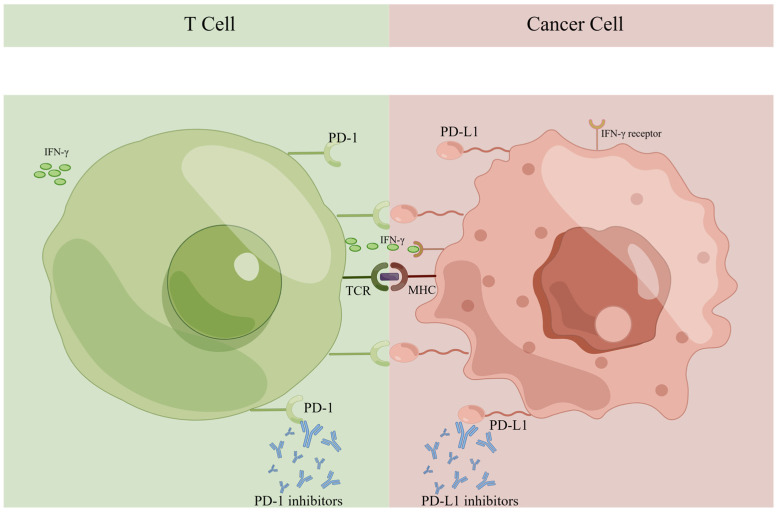
Immune checkpoints PD-1 and PD-L1 in the tumor microenvironment. PD-1, programmed cell death receptor 1; PD-L1, programmed cell death ligand 1; IFN-γ, interferon-γ; TCR, T-cell receptor; MHC, major histocompatibility complex.

**Table 1 jpm-13-00526-t001:** PD-1/PD-L1 inhibitors plus neoadjuvant chemotherapy trials in early-stage TNBC.

Trial	Phase	Enrolled Population	N	Regimen	Results	Status
GeparNuevo (NCT02685059)	II	Primary cT1b–cT4a-d TNBC irrespective of nodal involvement	174	Durvalumab + nab-paclitaxel vs. placebo + nab-paclitaxel	pCR: 53.4% vs. 44.2%; 3-year iDFS: 85.6% vs. 77.2%; 3-year DDFS: 91.7% vs. 78.4%; 3-year OS: 95.2% vs. 83.5%	Completed
I-SPY2 (NCT01042379)	II	HER2-negative early-stage breast cancer	250	Pembrolizumab + chemotherapy vs. chemotherapy	pCR in total: 44% vs. 17%; pCR in TNBC: 60% vs. 22%	Ongoing
NeoPACT (NCT03539948)	II	Stage I, II, or III TNBC	121	Pembrolizumab + chemotherapy	pCR: 58%; 24-month EFS: 89%	Ongoing
KEYNOTE-173 (NCT02622074)	Ib	Early-stage high-risk TNBC	60	Pembrolizumab + chemotherapy	pCR: 60%; 12-month OS: 80–100%	Completed
KEYNOTE-522 (NCT03036488)	III	Previously untreated, non-metastatic TNBC	1174	Pembrolizumab + chemotherapy vs. placebo + chemotherapy	pCR: 64.8% vs. 51.2%; 36-month EFS: 84.5% vs. 76.8%	Ongoing
NeoTRIPaPDL1 (NCT02620280)	III	Early-stage high-risk TNBC and locally advanced TNBC	278	Atezolizumab + chemotherapy vs. chemotherapy	pCR: 43.5% vs. 40.8%	Ongoing
IMpassion 031 (NCT03197935)	III	Early-stage TNBC	333	Atezolizumab + chemotherapy vs. placebo + chemotherapy	pCR in ITT population: 58% vs. 41%; pCR in PD-L1–positive population: 69% vs. 49%	Completed
NSABP B-59 (NCT03281954)	III	Clinical stage T2 or T3 TNBC	1550	Placebo + chemotherapy vs. atezolizumab + chemotherapy	NA	Ongoing
MIRINAE (NCT03756298)	II	TNBC patients with residual tumors after neoadjuvant chemotherapy	284	Atezolizumab + capecitabine vs. capecitabine monotherapy	NA	Ongoing

Abbreviations: pCR, pathological complete response; iDFS, invasive disease-free survival; DDFS, distant disease-free survival; OS, overall survival; EFS, event-free survival; ITT population, intention-to-treat population; TNBC, triple-negative breast cancer; PD-L1, programmed cell death ligand 1; vs., versus; NA, not available.

**Table 2 jpm-13-00526-t002:** PD-1/PD-L1 inhibitors plus adjuvant chemotherapy trials in early-stage TNBC.

Trial	Phase	Enrolled Population	N	Regimen	Results	Status
IMpassion 030 (NCT03498716)	III	Operable non-metastatic stage II or III TNBC	2300	Atezolizumab + anthracycline + paclitaxel vs. anthracycline + paclitaxel	NA	Ongoing
A-Brave (NCT02685059)	III	Non-metastatic TNBC	474	Avelumab vs. observation	NA	Ongoing
SWOG 1418 (NCT02954874)	III	Early-stage TNBC	1155	Pembrolizumab vs. observation	NA	Ongoing

Abbreviations: TNBC, triple-negative breast cancer; vs., versus; NA, not available.

**Table 3 jpm-13-00526-t003:** PD-1/PD-L1 inhibitors plus targeted therapy trials in early-stage TNBC.

Trial	Phase	Enrolled Population	N	Regimen	Results	Status
I-SPY2 (NCT01042379)	II	High-risk stage II or III HER2-negative breast cancer	372	Durvalumab + Olaparib + chemotherapy vs. chemotherapy	pCR in total: 37% vs. 20%; pCR in TNBC: 47% vs. 27%	Ongoing
NCT03594396	I/II	Stage II or III TNBC	54	Olaparib and durvalumab before neoadjuvant chemotherapy	NA	Ongoing
BARBICAN (NCT05498896)	II	Early-stage TNBC	146	Atezolizumab + chemotherapy vs. atezolizumab + ipatasertib + chemotherapy	pCR: 49.3% vs. 48.5%	Ongoing
BRE-03 (NCT04427293)	I	Stage I, II, or III TNBC	12	Lenvatinib + pembrolizumab	NA	Recruiting
NeoCAT (NCT05556200)	II	Operable invasive TNBC with TILs > 10%	58	Apatinib + camrelizumab	NA	Recruiting
NCT05112536	II	Early-stage TNBC	24	Trilaciclib + chemotherapy + pembrolizumab	NA	Ongoing

Abbreviations: pCR, pathological complete response; TNBC, triple-negative breast cancer; TILs, tumor-infiltrating lymphocytes; vs., versus; NA, not available.

## Data Availability

Not applicable.

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
