# Peer review of "Immunotherapy Targeting PD-1/PD-L1 in Early-Stage Triple-Negative Breast Cancer"

_jpm, 2023, doi:10.3390/jpm13030526_

Round 1
Reviewer 1 Report
General comments
- The title "Immunotherapy Targeting PD-1/PD-L1 in Early-stage Triple-Negative Breast Cance" is very interesting to better understand the role of PD-L1/PD-1 in triple negative breast cancer.
- The authors summarize the potential combination of immunotherapies and chemotherapies in triple negative breast cancer, which gives concise information about the role PD-L1/PD-1 inhibitors.
-However, this study is somewhat narrow and not as in-depth as a review article, does not demonstrate any particular innovation of the study, and is difficult to publish as it is.
- There are also more thorough meta-analyses and systematic reviews related to the topic. Here are some examples.
1. Emi Noguchi, Tadahiko Shien, Hiroji Iwata, Current status of PD-1/PD-L1 blockade immunotherapy in breast cancer, Japanese Journal of Clinical Oncology, Volume 51, Issue 3, March 2021, Pages 321–332, https://doi.org/10.1093/jjco/hyaa230
2. Zhang B, Liu Y, Zhou S, Jiang H, Zhu K, Wang R. Predictive effect of PD-L1 expression for immune checkpoint inhibitor (PD-1/PD-L1 inhibitors) treatment for non-small cell lung cancer: A meta-analysis. Int Immunopharmacol. 2020 Mar;80:106214. doi: 10.1016/j.intimp.2020.106214. Epub 2020 Jan 23. PMID: 31982822.)
(2022) Atezolizumab and pembrolizumab in triple-negative breast cancer: a meta-analysis, Expert Review of Anticancer Therapy, 22:2, 229-235, DOI: 10.1080/14737140.2022.2023011
Author Response
Dear reviewer,
We feel great thanks for your professional comments concerning our manuscript “Immunotherapy Targeting PD-1/PD-L1 in Early-stage Triple-Negative Breast Cancer”. Those comments are all valuable and very helpful for furthermore improving our paper. We read the comments carefully and have made corresponding corrections which we hope to meet with your approval.
Yours sincerely,
Tinglin Yang, Wenhui Li, Tao Huang, and Jun Zhou
The main corrections in the paper and the response to your comments are as follows. All modifications in the manuscript have been marked up by using the “track changes” function in MS Word.
Comment:
General comments
- The title "Immunotherapy Targeting PD-1/PD-L1 in Early-stage Triple-Negative Breast Cance" is very interesting to better understand the role of PD-L1/PD-1 in triple negative breast cancer.
- The authors summarize the potential combination of immunotherapies and chemotherapies in triple negative breast cancer, which gives concise information about the role PD-L1/PD-1 inhibitors.
Response:
Thanks for your suggestions and the suggestion is appreciated.
Comment:
-However, this study is somewhat narrow and not as in-depth as a review article, does not demonstrate any particular innovation of the study, and is difficult to publish as it is.
- There are also more thorough meta-analyses and systematic reviews related to the topic. Here are some examples.
- Emi Noguchi, Tadahiko Shien, Hiroji Iwata, Current status of PD-1/PD-L1 blockade immunotherapy in breast cancer, Japanese Journal of Clinical Oncology, Volume 51, Issue 3, March 2021, Pages 321–332, https://doi.org/10.1093/jjco/hyaa230
- Zhang B, Liu Y, Zhou S, Jiang H, Zhu K, Wang R. Predictive effect of PD-L1 expression for immune checkpoint inhibitor (PD-1/PD-L1 inhibitors) treatment for non-small cell lung cancer: A meta-analysis. Int Immunopharmacol. 2020 Mar;80:106214. doi: 10.1016/j.intimp.2020.106214. Epub 2020 Jan 23. PMID: 31982822.)
- Farah Latif, Hira Bint Abdul Jabbar, Hamna Malik, Humaira Sadaf, Azza Sarfraz, Zouina Sarfraz & Ivan Cherrez-Ojeda (2022) Atezolizumab and pembrolizumab in triple-negative breast cancer: a meta-analysis, Expert Review of Anticancer Therapy, 22:2, 229-235, DOI: 10.1080/14737140.2022.2023011
Response:
In this review, we provide clinicians and patients with the latest data regarding PD-1/PD-L1 inhibitors in early-stage TNBC, which to our knowledge has never been systematically reviewed.
Based on the comments, we concluded and quoted the reviews and meta-analyses mentioned in the comment in the section Introduction (page 2, line 62).
Details of progresses and innovations in our review are listed as follows, distinguishing it from previous studies.
- The review “Current status of PD-1/PD-L1 blockade immunotherapy in breast cancer” published in 2021 mainly included clinical trials in metastatic TNBC. There were 6 clinical trials combining immunotherapy with chemotherapy in early-stage TNBC, but results were incomplete due to the uncompleted status of clinical trials by the published time. Up to now, numerous new results have emerged, including the pCR rates in I-SPY2 trail, the 36-month EFS rates in KEYNOTE-522 trial, the pCR rates in IM passion 031 trial, and encouraging 3-year iDFS rates, DDFS rates, and OS rates in GeparNuevo trial, etc. There was also a trend attempting to combine immunotherapy with chemotherapy and various types of targeted therapy for early-stage TNBC patients, as 12 more trials were reviewed in sections 3,4, and 5 of our review.
- The meta-analysis “Predictive effect of PD-L1 expression for immune checkpoint inhibitor (PD-1/PD-L1 inhibitors) treatment for non-small cell lung cancer: A meta-analysis” analyzed 6 studies of ICIs in advanced NSCLC patients. Whether the conclusion for NSCLC can be applicable to early-stage TNBC patients needs more discussion and exploration. In fact, PD-L1 can be used as a potential biomarker for early-stage TNBC patients to some extent. Other evidence from TNBC trials and 3 more potential biomarkers were discussed in section 6 of our review.
- The meta-analysis “Atezolizumab and pembrolizumab in triple-negative breast cancer: a meta-analysis” analyzed the efficacy of PD-1/PD-L1 inhibitors in 6 neoadjuvant or adjuvant trials. But the efficacy of anti-PD-1/PD-L1 therapy in early-stage TNBC was not concluded in particular.
Therefore, it is necessary to review the latest data from clinical trials comprehensively and summarize the role of PD-1/PD-L1 inhibitors, especially in early-stage TNBC. Challenges facing the field were also concluded in section 8 of our review, indicating the efforts and future trends to improve immunotherapy.
Reviewer 2 Report
In the current manuscript, Yang T, et al. summarized current clinical trials of immunotherapy targeting PD-1/PD-L1 in early stage TNBC. The manuscript will be of interest for readers in breast cancer biology field. Specific comments are:
1. The meaning of some language or terms used are unclear or ambiguous at best. Professional proofreading and editing service is strongly recommended.
2. The full name of acronym should be spelled out when it first appears in the text, including irAEs.
3. Table 2 can be divided into 2 tables, one for the PD-1/PD-L1 inhibitors plus adjuvant chemotherapy, and the other for PD-1/PD-L1 inhibitors plus targeted therapy.
4. It is suggested that a section that describes challenges be added to provide a better understanding into the efforts needed to improve immunotherapy in TNBC.
Author Response
Dear reviewer,
We feel great thanks for your professional comments concerning our manuscript “Immunotherapy Targeting PD-1/PD-L1 in Early-stage Triple-Negative Breast Cancer”. Those comments are all valuable and very helpful for furthermore improving our paper. We read the comments carefully and have made corresponding corrections which we hope to meet with your approval.
Yours sincerely,
Tinglin Yang, Wenhui Li, Tao Huang, and Jun Zhou
The main corrections in the paper and the response to your comments are as follows. All modifications in the manuscript have been marked up by using the “track changes” function in MS Word.
General comment: In the current manuscript, Yang T, et al. summarized current clinical trials of immunotherapy targeting PD-1/PD-L1 in early-stage TNBC. The manuscript will be of interest for readers in the breast cancer biology field. Specific comments are:
Response: Thanks for your valuable comments. We have taken all these comments into account, and have made major revisions to the resubmitted manuscript. Point-by-point responses are as follows.
Comment [1]: The meaning of some language or terms used are unclear or ambiguous at best. Professional proofreading and editing service is strongly recommended.
Response [1]: We revised the whole manuscript carefully to avoid typos and tried our best to improve the manuscript. In addition, the manuscript has been revised by a professional language editing service from MDPI to improve its readability. The language certificate is uploaded in the attachment, and we hope the revised manuscript could be acceptable.
Comment [2]: The full name of acronym should be spelled out when it first appears in the text, including irAEs.
Response [2]: We sincerely thank the reviewer for the careful reading. As suggested, we checked all acronyms and their full names. For the acronym “irAEs”, we spelled out its full name “immune-related adverse events” in the section Introduction (page 2, line 68), thus the acronym was used in section 7 irAEs (page 10, line 405).
Comment [3]: Table 2 can be divided into 2 tables, one for the PD-1/PD-L1 inhibitors plus adjuvant chemotherapy, and the other for PD-1/PD-L1 inhibitors plus targeted therapy.
Response [3]: Based on the suggestion, we divided Table 2 into two tables: Table 2 for PD-1/PD-L1 inhibitors plus adjuvant chemotherapy (page 7, line 261), and Table 3 for PD-1/PD-L1 inhibitors plus targeted therapy (page 8, line 348).
Comment [4]: It is suggested that a section that describes challenges be added to provide a better understanding of the efforts needed to improve immunotherapy in TNBC.
Response [4]: Potential directions for future efforts are indeed a crucial part of improving immunotherapy. We have added section 8 Challenges in TNBC Immunotherapy (page 11, lines 453). We also gave a brief summarization of this part in the sections including Abstract (page 1, line 19), Introduction (page 2, line 69), and Conclusions (page 12, line 507).

Round 2
Reviewer 1 Report
Minor comments
- As a review paper, the author should cite references that were published within the last five years. There are some references dates back more than five years.
Author Response
Dear reviewer,
We feel great thanks for your professional comment concerning our manuscript “Immunotherapy Targeting PD-1/PD-L1 in Early-stage Triple-Negative Breast Cancer”. The comment is valuable and very helpful for furthermore improving our paper. We read the comment carefully and have made corresponding corrections which we hope to meet with your approval.
Yours sincerely,
Tinglin Yang, Wenhui Li, Tao Huang, and Jun Zhou
The main corrections in the paper and the response to the reviewers’ comment are as follows. All modifications in the manuscript have been marked up by using the “track changes” function in MS Word.
Comment: Minor comments
- As a review paper, the author should cite references that were published within the last five years. There are some references dates back more than five years.
Response: As suggested, we have checked the references carefully. A total of 17 references within the last five years were re-cited, without influencing the intended meaning of our review. Despite the few references we quoted were not the latest, those articles were hot and significant papers in the field of early-stage TNBC immunotherapy, which was consistent with the main content of our review. We sincerely hope the cited references could be acceptable to you and thank you again for your careful reading.